# Scalable and Accurate Graph Reasoning with LLM-based Multi-Agents

## Abstract

Recent research has explored the use of Large Language Models (LLMs) for tackling complex graph reasoning tasks. However, due to the intricacies of graph structures and the inherent limitations of LLMs in handling long text, current approaches often fail to deliver satisfactory accuracy, even on small-scale graphs and simple tasks. To address these challenges, we introduce GraphAgent-Reasoner, a fine-tuning-free framework that utilizes a multi-agent collaboration strategy for explicit and precise graph reasoning. Inspired by distributed graph computation theory, our framework decomposes graph problems into smaller, node-centric tasks that are distributed among multiple agents. The agents collaborate to solve the overall problem, significantly reducing the amount of information and complexity handled by a single LLM, thus enhancing the accuracy of graph reasoning. By simply increasing the number of agents, GraphAgent-Reasoner can efficiently scale to accommodate larger graphs with over 1,000 nodes. Evaluated on the GraphInstruct dataset, our framework demonstrates near-perfect accuracy on polynomial-time graph reasoning tasks, significantly outperforming the best available models, both closed-source and fine-tuned open-source variants. Our framework also demonstrates the capability to handle real-world graph reasoning applications such as webpage importance analysis.

## 1 Introduction

Graphs, as a crucial data structure for modeling complex real-world relationships, are ubiquitous across various scenarios, *e.g.* citation networks, recommendation networks. Many important applications like drug discovery (Stokes et al., 2020), traffic forecasting (Jiang & Luo, 2022), and financial detection (Motie & Raahemi, 2024), require reasoning over graphs to be realized. Noticing the powerful general knowledge and language processing capabilities of Large Language Models (LLMs) (Brown et al., 2020), a significant amount of works have focused on using LLMs to perform various reasoning tasks, such as mathematical formula derivation (Meadows et al., 2023), commonsense reasoning (Madaan et al., 2022), and multi-hop question answering (Creswell et al., 2023). However, most of them primarily involve shallow or sequential reasoning. To bring the LLM reasoning closer to human thinking, it is necessary for LLMs to master deeper and more complex reasoning, such as graph reasoning.

Despite significant efforts by researchers to enable LLMs to memorize, comprehend, and perform basic reasoning on graph structures, several issues still persist: **1) The scale of graphs that can be handled is limited.** Describing graph structures in natural language inevitably leads to excessively long inputs. Due to context length limitations and the shortcomings of LLMs in handling lengthy text (Liu et al., 2023), previous works (Chai et al., 2023; Fatemi et al., 2024; Perozzi et al., 2024) could only handle graphs of very limited size (e.g. fewer than 20 nodes and 100 edges). **2) The performance on graph reasoning tasks is relatively poor.** Unlike text, which can tolerate some degree of semantic deviation, reasoning and computation on graphs must be highly precise. However, current works demonstrate poor accuracy (average 20~60%) in various graph reasoning tasks like connectivity and shortest path. **3) Lacking explicit reasoning paths.** Taking the shortest path as an example, the responses of existing models resemble a heuristic search approach to finding the shortest path on a graph, rather than strictly executing an algorithm. This makes it difficult to determine whether LLMs are genuinely deriving the answer through correct reasoning or merely making educated guesses. Although GraphWiz (Chen et al., 2024a) attempts to generate explicit reasoning

paths through fine-tuning, it often fails due to the presence of incomplete or wrong reasoning paths in its training data. Furthermore, GraphWiz exhibits overfitting, where it tends to treat new or unrelated questions as one of the fine-tuned problems, which will be detailed in Section 5.3.

**Motivation.** The ultimate goal of graph reasoning is to enable LLMs to leverage graph-related knowledge or algorithms to solve real-world graph problems. However, with the development of information science and hardware storage, the scale of graphs and information per node become too large for a single LLM to handle. To address this, a natural idea is to use distributed approaches, where a large graph is stored across multiple LLMs separately and compute collaboratively. Therefore, just as graph algorithms have generally evolved from non-distributed to distributed forms (Meng et al., 2024b)), we hope that LLMs can also learn the concept of distributed processing, thereby harnessing the power of swarm intelligence to solve graph problems in real-world scenarios.

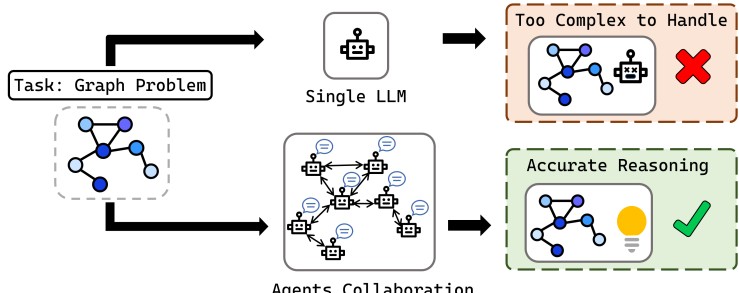

Figure 1: The current situation of LLMs in solving graph problems. Previous methods using a single LLM often failed due to the complex graph structures. In contrast, our approach leverages agents collaboration to effectively address graph problems.

**Our Contribution.** To address the above limitations, in this paper, we propose the GraphAgent-Reasoner(GAR) framework, which leverages the power of swarm intelligence to solve graph reasoning problems, as shown in Figure 1. We follow a node-centric approach, assigning an agent to each node, allowing it to focus on processing its own information and communicate with neighbors. Thus, we can easily scale up the size of graphs that can be processed by simply increasing the number of agents. At the same time, under the direction of a Master LLM, graph problems are decomposed into smaller, node-centric tasks, which are assigned to agents for collaborative resolution. This approach significantly reduces the scale and complexity of information each agent needs to process, thereby greatly improving the overall accuracy. Furthermore, since agents must clearly transmit the processed information to neighboring agents, the reasoning process becomes transparent, demonstrating the framework solves graph reasoning problems through clear and correct reasoning, rather than lucky guessing. In summary, our contributions are as follows:

- We propose GraphAgent-Reasoner, the first LLM-based multi-agents framework for graph reasoning, which requires no fine-tuning and can utilize any LLM as the underlying reasoning model. Our framework achieves near-perfect accuracy on various polynomial-time tasks, significantly surpassing the performance of existing methods.

- Our framework expands the scale of graph reasoning tasks handled by LLMs from 100 nodes to 1,000 nodes, demonstrating exceptional scalability. Furthermore, as the graph size increases, our framework does not exhibit the significant performance degradation seen in other methods and maintains robust accuracy.

- We explore the performance of our framework in real-world applications like webpage importance analysis, showcasing its potential for addressing complex graph reasoning problems in real-life situations.

## 2 PRELIMINARIES AND RELATED WORKS

**Preliminaries.** In general scenarios, when discussing LLMs solving graph reasoning problems, the input is a $(\mathcal{G}, \mathcal{Q})$ pair. $\mathcal{G}$ is a graph represented as $\mathcal{G} = (\mathcal{V}, \mathcal{E}, \{s_i\}, \{t_i\})$, where $\mathcal{V}$ is the node set and $\mathcal{E}$, the edge set. For each node $v_i \in \mathcal{V}$, a sequential text node feature $s_i$ is associated; similarly, for

each edge $e_i \in \mathcal{E}$, a sequential text edge feature $t_i$ is assigned. The graph $\mathcal{G}$ is described in natural language, typically using edge or adjacency list representation. $\mathcal{Q}$ is a task-specific instruction or problem description. LLMs will process the $(\mathcal{G}, \mathcal{Q})$ pair and return an answer string $A$.

**Large Language Models for Graph Reasoning.** To further enhance the reasoning capabilities of LLMs, many works have attempted to improve the performance of LLMs in graph reasoning. Wang et al. (2023) first introduces the NLGraph Benchmark to evaluate the performance of LLMs on various graph reasoning tasks. Fatemi et al. (2024) explores the impact of different graph encoding methods and graph structure types on the performance of LLMs in graph reasoning tasks. Additionally, it introduces another benchmark called GraphQA. Considering the lengthy nature of describing graph structures in text, Chai et al. (2023) and Perozzi et al. (2024) respectively use Transformers and GNNs to encode graph structures and attempt to align them with LLMs. Inspired by how humans understand structural information through the visual modality, Wei et al. (2024) generates corresponding visual images based on graph structures and provides them to visual LLMs for graph reasoning. Chen et al. (2024a) conducted Supervised Fine-Tuning and Directly Prefered Optimization on LLMs, enhancing the performance of LLMs and encouraging them to output explicit reasoning paths.

**Large Language Model based Multi-Agents.** Recent advancements in LLMs have spurred interest in their application within multi-agent systems. LLM-based multi-agent frameworks leverage the natural language understanding and reasoning capabilities of LLMs to enable agents to collaborate, communicate, and solve complex tasks in a distributed manner. Existing multi-agents works for problem solving primarily focuses on applications such as Software Development (Dong et al., 2023; Hong et al., 2024; Qian et al., 2024), Embodied Agents (Zhang et al., 2024; Mandi et al., 2024; Chen et al., 2024b) and Science Debate (Xiong et al., 2023; Chan et al., 2024). However, using LLM-based multi-agents to handle graph data has been less explored, especially in the areas of graph reasoning and graph computation tasks. This may be due to the hallucination issue inherent in LLMs (Huang et al., 2023), where their responses are factually incorrect. This problem becomes more complex in a multi-agent setting, as the hallucinations of a single agent may propagate to other nodes by communication (Guo et al., 2024). This requires the performance of individual agents be sufficiently stable to ensure the correct operation of the entire multi-agent system.

# 3 LIMITATIONS OF SINGLE LLM IN GRAPH REASONING

Although LLMs exhibit strong language processing and logical reasoning capabilities, problems with the Transformer architecture and Attention mechanism (Vaswani et al., 2017) still limit the scale and accuracy when they process graph problems. There are two primary limitations:

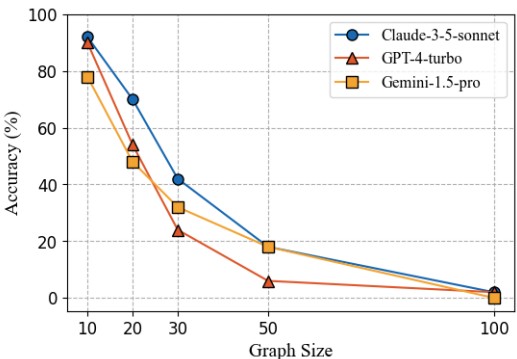

Figure 2: The performance of a single LLM in memorizing first-order neighboring nodes. As the number of nodes increases, all models exhibit significant memory errors.

**The graph structure is too complex to memorize and understand for a single LLM.** Using adjacency or edge lists to describe graph structures in natural language is the most intuitive and direct method, facilitating the processing of graph data by LLMs through text. However, this approach inevitably leads to a lengthy context, as the number of edges can grow quadratically with the number of nodes. As the graph scales up and becomes denser, the graph structure becomes highly complex, requiring a large amount of tokens to describe the edge relationships. When the text becomes too

lengthy, it becomes difficult for LLMs to properly allocate attention, and they may even struggle with simple tasks such as key-value pair matching Liu et al. (2023). This presents significant challenges for LLMs in identifying key information for graph reasoning tasks from the lengthy context. Figure 2 shows the performance of a single LLM in memorizing one-hop neighbor nodes. We observe that as the number of nodes in the graph increases, various LLMs exhibit a significant decline in accuracy. If a single LLM cannot even correctly recall basic graph structural information like node neighbors, it becomes difficult to proceed with more complex graph reasoning or computation.

Furthermore, the graph structure is described in a sequential manner. LLMs have to identify implicit graph structures from sequential text. Since the processing of LLMs is a black-box operation, it is difficult to assert that they truly construct graph structures implicitly and thereby understand them. Huang et al. (2024) conducted extensive experiments to explore whether LLMs treat the input prompts as graphs or merely as paragraphs with keywords on TAGs. The results show that the performance of LLMs in handling TAGs primarily stems from the context rather than the graph structure. LLMs tend to process the graph description as linearized paragraphs rather than graphs.

**A single LLM struggles to solve reasoning problems in real-world scenarios.** Researchers train LLMs on graph reasoning tasks to empower them to utilize learned graph-related knowledge or algorithms to tackle real-world graph problems. However, in practical scenarios, the amount of information associated with each node can be enormous. Take citation networks as an example: a single node represents a paper, and its node information includes the title, abstract, and references, which could amount to several thousand tokens. In addition to the complexity of graph structures, the need to handle a large amount of node information further exacerbates the burden on a single LLM and highlights its shortcomings in processing long contexts. Moreover, using a single LLM to handle the entire network is inefficient, as it cannot coherently process the entire network's problems. Typically, it is necessary to manually compress or summarize the information for each node and then feed local subgraphs to the LLM for processing (Guo et al., 2023; Chen et al., 2023).

Furthermore, many current works (Chen et al., 2024a; Perozzi et al., 2024) require training GNNs or fine-tuning LLMs on individual or multiple graph reasoning tasks. However, when transferring to other graph tasks, a certain degree of performance degradation occurs, and retraining or fine-tuning for new graph tasks consumes a significant amount of time and resources. Whether LLMs can apply the graph knowledge and algorithms learned during the training process to actual graph reasoning also remains an open question. We explored this question in 5.3 and observed significant overfitting in LLMs fine-tuned on specific graph reasoning tasks. Therefore, the ideal solution would be to leverage the powerful general knowledge acquired during the pre-training phase of LLMs through an appropriate approach, enabling them to handle graph reasoning tasks as naturally as they do with natural language problems.

# 4  GRAPHAGENT-REASONER

To solve the limitations above, we propose a novel framework based on multi-agent collaboration called GraphAgent-Reasoner as shown in Figure 3, aiming to solve graph reasoning problems explicitly and correctly. The interface of the framework is a Master LLM, which is responsible for processing the textual input of graph problems, constructing the agent network, directing them to collaboratively solve the problem, and finally aggregating the states of all agents to derive the solution. Its implementation is based on the React Agent proposed by Yao et al. (2023), which is capable of reasoning based on the environment and executing corresponding actions, as detailed later. The pipeline of GAR consists of four steps: Graph Construction, Algorithm Establishing, Distributed Execution and Master Summarization.

**Graph Construction.** Given an input pair $(\mathcal{G}, \mathcal{Q})$, the Master LLM first extracts the node and edge information from the textual description of graph $\mathcal{G}$. It then constructs an agent for each node and initializes the node's state and neighbor information, forming an interconnected network of agents. Each agent independently maintains its state and neighbor data, communicates with adjacent agents based on instructions from the Master LLM, and updates its state in each round.

**Algorithm Establishing.** To accommodate diverse graph tasks and fully exploit the knowledge embedded in LLMs during pre-training, we propose a unified solution approach framed within a

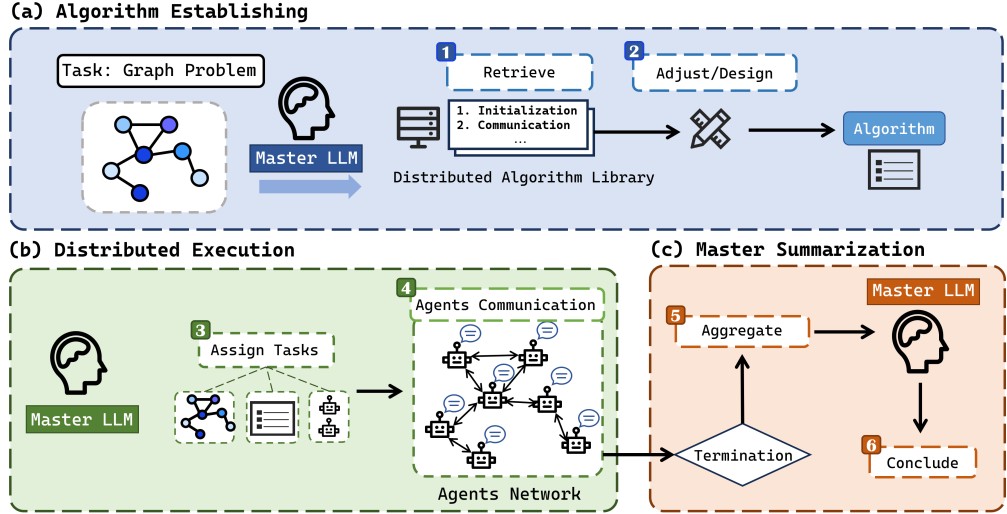

Figure 3: The framework of GraphAgent-Reasoner. Given a graph problem, the Master LLM will first construct agents network according to graph strcutures. It then sequentially performs Algorithm Establishing, Distributed Execution and Master Summarization, as detailed in this section.

distributed paradigm as shown in Algorithm 1. This approach requires the Master LLM to specify six core components for each problem: State, Message, Initialization, Send, Update, and Termination.

- **State**: The local information maintained by each node, representing its current state. This can include attributes like node features, labels, or any other task-specific data. The states evolve as nodes receive messages and update their information.

- **Message**: The data transmitted between nodes during the communication phase. Messages typically contain information that neighboring nodes need to perform updates, such as feature values, distances, or other task-relevant information.

- **Initialization**: At the start of the execution, each node initializes its state with predefined values, which may be based on node IDs, input features or task-specific requirements. This step ensures that the graph is ready to begin the communication process.

- **Send**: After initialization, each node generates messages based on its current state and sends them to its neighboring nodes. This step is repeated in each iteration, allowing nodes to continuously exchange information with their neighbors.

- **Update**: Upon receiving messages from its neighbors, each node updates its state by aggregating the incoming messages and combining them with its current state. This iterative process enables nodes to refine their information over time.

- **Termination**: The algorithm halts when a predefined stopping condition is met, such as reaching a fixed number of iterations, achieving convergence, or satisfying a task-specific criterion. Once the termination condition is reached, each node will send its final state to the Master LLM, and the execution terminates.

Since LLMs lack prior knowledge of this distributed paradigm, to facilitate the Master LLM's understanding and application of the framework, we develop a distributed algorithm library that adheres to this distributed paradigm, from which the Master LLM can query relevant algorithm templates to generate distributed solutions within this paradigm. Specifically, we selected classic distributed graph algorithms and documented their implementations under this distributed paradigm. Some examples are presented in Appendix A.1. Drawing on prior work (Zheng et al., 2024; Meng et al., 2024a), we endeavor to write detailed reasoning steps of each part in the algorithm to encourage the agent to think step by step as much as possible, which plays an important role in enhancing the success rate of individual agents.

When receiving a problem input, the Master LLM first retrieves the $k$ algorithms most relevant to the problem description from the distributed algorithm library. If there are algorithms suitable for

---

**Algorithm 1:** Distributed Paradigm

---

1 **Input**: Agent Nodes $\mathcal{A}$, each agent $a \in \mathcal{A}$ maintains a state $S_a$, the maximum iterations $I_{max}$ given by the Master LLM.
2 **Output**: Final state $S_a$ for each agent $a \in \mathcal{A}$
  /* Initialization */
3 Each agent $a \in \mathcal{A}$ initializes its state $S_a$ based on **Initialization** rules.
4 Each agent $a$ sends an initial message $M_{a \to v}$ to each of its neighbors $v \in$ Neighbors$(a)$ based on its current state $S_a$ and **Send** rules.
  /* Communication */
5 **while** *Iteration $i < I_{max}$ and **Termination** not met* **do**
6     a. /* Receive */
7     Each agent $a$ receives messages $M_{u \to a}$ from all neighboring agents $u$.
8     b. /* Update */
9     Each agent $a$ updates its state $S_a$ based on the received messages $M$ and its own current state $S_a$ according to **Update** rules.
10    c. /* Send */
11    Each agent $a$ sends updated messages $M_{a \to v}$ to each of its neighbors $v$ based on the updated state $S_a$ according to **Send** rules.
12 **Return**: the final state $S_a$ for all agents $a \in \mathcal{A}$

---

handling the problem, the Master LLM will adjust the algorithm according to the problem description, such as changing the initialization and termination conditions (e.g., the source node in the shortest path problem). If there are no appropriate algorithms, the Master LLM will design a distributed algorithm following the distributed paradigm based on the examples of the retrieved algorithms. For some generated examples, see Appendix A.2.

**Distributed Execution.** After the distributed algorithm is designed, the Master LLM will relay the approach to each agent node for execution according to the process outlined in Algorithm 1. Each agent will first initialize its state based on node information and algorithm rules and then send an initial message to neighboring agents. Subsequently, each agent will iteratively execute the operations of receiving messages, updating its state, and sending messages according to the algorithm rules, synchronizing progress after each communication round. Communication will continue until the maximum number of iterations is reached or the termination condition is met.

**Master Summarization.** Finally, the final state of all agent nodes will be aggregated to the Master LLM, which will summarize the results conclude based on the problem and return the final answer in natural language form.

## 5 EXPERIMENTS

In this section, we summarize the key experiments conducted with GAR. We begin by highlighting some of the most exciting results from our analysis here:

- **R1**: GAR achieves **near-perfect accuracy** on polynomial-time graph reasoning problems, significantly surpassing existing closed-source models and open-source models fine-tuned on extensive data.

- **R2**: GAR maintains high accuracy on larger-scale graphs (**up to 1000 nodes**), demonstrating superior scalability. In contrast, as the number of nodes increases, other models exhibit a significant decline in performance or become incapable of handling the problem at all due to the context length limitation.

- **R3**: GAR showcases a robust understanding and application of graph algorithms in real-world graph reasoning scenarios, highlighting its potential for addressing complex graph problems encountered in daily life. In contrast, other open-source models that have undergone extensive fine-tuning on graph reasoning datasets fail to apply the learned graph reasoning knowledge when confronted with rephrased real-world graph problems.

**Datasets.** We conduct our experiments on the graph reasoning tasks proposed in GraphInstruct (Chen et al., 2024a). This dataset contains nine graph reasoning problems with different time complexity, ranging from linear and polynomial complexity to NP-complete.

- **Linear.** Cycle Detection (Detect if a given graph $\mathcal{G}$ contains any cycles), Connectivity (Assess if two nodes $u$ and $v$ in a given graph $\mathcal{G}$ are connected via a path), Bipartite Graph Check (Judge if a given graph $\mathcal{G}$ is bipartite), and Topological Sort (Find a topological ordering of vertices in a directed acyclic graph $\mathcal{G}$).
- **Polynomial.** Shortest Path (Compute the shortest path between two specific nodes $u$ and $v$ in a given graph $\mathcal{G}$), Maximum Triangle Sum (Find the maximum sum of weights for any connected triplet of vertices in a given graph $\mathcal{G}$), and Maximum Flow (Calculate the maximum flow from a source node $s$ to a sink node $t$ in a directed graph $\mathcal{G}$).

Due to the complexity of NP-complete problems, there are currently no mature exact distributed algorithms available for their solution. Consequently, the Master LLM is unable to design correct and effective distributed algorithms based on the knowledge acquired during pre-training. Therefore, in our experiments, we only consider linear and polynomial-time problems. Detailed information of the dataset and partial test results for NP-complete problems will be presented in Appendix B.

**Setting.** The underlying reasoning LLM of Agent Node used in our framework is ChatGPT-4o-mini-2024-07-18, and the base model of Master LLM is ChatGPT-4-turbo(OpenAI, 2023). The temperature is consistently set to 0. Our framwork is built upon AgentScope (Gao et al. (2024)), an innovative platform to easily build reliable, high-performance multi-agent applications.

## 5.1 EXPERIMENT 1: PERFORMANCE ON GRAPHINSTRUCT

In this experiment, we evaluate the performance of GAR on polynomial-time tasks of the GraphInstruct dataset. The results are shown in Table 1. We see GAR exhibits near-perfect results on these tasks, significantly outperforming other models. Especially on shortest and triangle tasks with high time complexity, GAR substantially improves the performance of LLMs. Problems that a single LLM struggles to solve have been effectively resolved through collaboration by agents after being decomposed into smaller, node-centric tasks.

As the number of nodes increases, the graph structures become more complex, making the solution of graph problems increasingly difficult. To investigate how the performance of models varies with increasing problem complexity, we conduct experiments on cycle detection and shortest path problems, gradually increasing the number of nodes from 5 to 100. The results are presented in Figure 4.

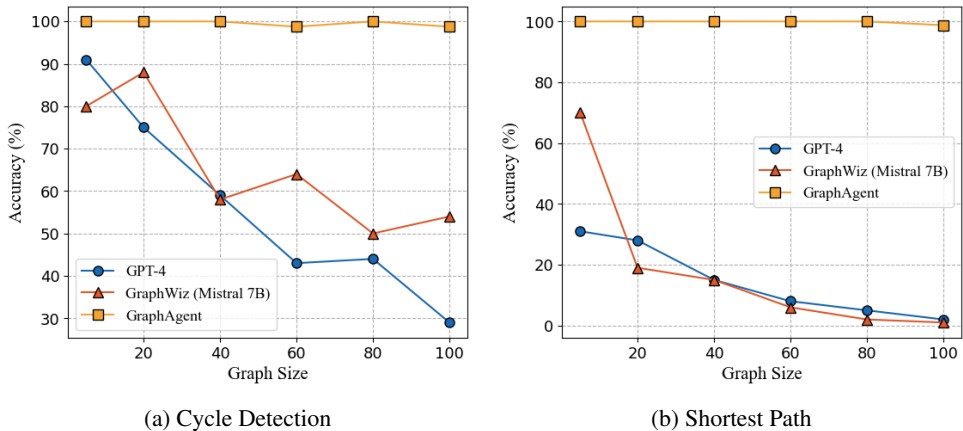

(a) Cycle Detection         (b) Shortest Path

Figure 4: Performance of GraphAgent-Reasoner, GPT4(2 shot) and GraphWiz(Mistral 7B) on cycle detection and shortest path problems with different graph sizes.

We see with the number of nodes increasing, both ChatGPT-4 and Graphwiz exhibit a significant decline in performance. However, the accuracy of GAR remains stable, almost unaffected by the graph size, demonstrating robust scalability. Although the scale of the graph is increasing, the

information processed by each agent has not significantly increased. Each agent still only handles its own information and communicates with neighboring agents. We observe that GAR occasionally makes errors in specific cases, likely due to the increasing communication rounds as the number of nodes and edges grows. Even when handling simple node-centric tasks, a single agent still has the potential to make mistakes. Therefore, as the number of agents and communication rounds increases, the overall likelihood of errors also rises. This can be improved by enhancing the capability of individual agents (such as using stronger LLMs as the underlying reasoning model) or by more finely designed prompts.

Table 1: Performance of GraphAgent-Reasoner and other models on polynomial-time tasks of GraphInstruct test set. Each task contains 400 test cases, with a maximum of 100 nodes. The first best result for each task is highlighted in bold, and the second best result is highlighted underlined.

| Models | Linear | | | | Polynomial | | Average |
|---|---|---|---|---|---|---|---|
| | cycle | connect | bipartite | topology | shortest | triangle | |
| **Closed-source Models** | | | | | | | |
| GPT-4 (zero-shot) | 38.75 | 17.00 | 65.25 | 5.00 | 9.25 | 5.75 | 23.50 |
| GhatGPT (2-shot) | 51.25 | 43.75 | 70.75 | 4.50 | 3.50 | 17.25 | 31.83 |
| GPT-4 (2-shot) | 52.50 | 62.75 | 74.25 | 25.25 | 18.25 | 31.00 | 44.00 |
| **Fine-tuned Open-source Models** | | | | | | | |
| Naive SFT (LLaMA 2-7B) | 73.75 | 83.50 | 41.25 | 4.00 | 9.50 | 30.00 | 40.17 |
| Naive SFT (Mistral-7B) | 73.75 | 83.50 | 78.50 | 1.00 | 23.00 | 47.00 | 51.13 |
| GraphWiz (LLaMA 2-7B) | 91.50 | 87.00 | 74.00 | 18.00 | 28.00 | 38.25 | 56.13 |
| GraphWiz (Mistral-7B) | 92.00 | 89.50 | 72.00 | 19.00 | 31.25 | 38.75 | 57.08 |
| GraphWiz-DPO (LLaMA 2-7B) | 89.00 | 82.50 | 84.75 | 46.75 | 24.00 | 52.75 | 63.29 |
| GraphWiz-DPO (Mistral-7B) | 85.50 | 79.50 | 85.50 | 85.25 | 12.50 | 29.00 | 62.88 |
| GraphAgent-Reasoner | **99.50** | **100.00** | **100.00** | **96.50** | **99.75** | **93.25** | **98.00** |

## 5.2 EXPERIMENT 2: PERFORMANCE ON LARGE-SCALE GRAPHS

In this experiment, we evaluate the performance of current LLMs on large-scale graphs. The largest graph size handled by existing graph reasoning work is 100 nodes (Chen et al., 2024a), which is still far from sufficient for real-world graph reasoning scenarios. To evaluate the reasoning performance of existing models on larger graphs, we conduct shortest path experiments on graphs with 100, 200, 500, and 1000 nodes. Due to the excessively long input text (reaching 16,000 tokens for 1000 nodes) and the money cost, we only create 20 test samples for each graph size. The results are shown in Table 2.

Table 2: Performance on large-scale graphs dealing with shortest path problems. x/20 indicates that out of 20 test samples, x samples are correct. NA signifies that testing could not be conducted due to the fact that the context length limit is exceeded.

| Graph Size | 100 | 200 | 500 | 1000 |
|---|---|---|---|---|
| Graphwiz (LLaMA 2-7B) | 0/20 | 0/20 | NA | NA |
| Graphwiz (LLaMA 2-7B-DPO) | 0/20 | 0/20 | NA | NA |
| Chatgpt-3.5-turbo-16k | 0/20 | 0/20 | 0/20 | 0/20 |
| Chatgpt-4-32k | 0/20 | 1/20 | 0/20 | 0/20 |
| GraphAgent-Reasoner | **20/20** | **20/20** | **20/20** | **18/20** |

We see the two GraphWiz models fine-tuned on the LLaMA2-7B (Touvron et al., 2023) base model are unable to handle graphs with 500 or more nodes due to the context length limitation (the context length limit for Llama2 is 4096 tokens). Although ChatGPT-3.5-turbo-16k and ChatGPT-4-32k can manage longer contexts, they output wrong answers in almost all test samples, with only ChatGPT-4-32k being correct in one 200 nodes test sample. In contrast, GAR maintains a high accuracy in large-scale graph, only failed in two 1000-node test samples, further demonstrating its robust scalability.

## 5.3 EXPERIMENT 3: CASE STUDY

In this experiment, we explore the application of two graph reasoning models, Graphwiz and GAR, in real-world graph reasoning scenarios. We present a case study of webpage importance analysis in Figure 5.

```
(G,Q): For a webpage network graph, each node represents a webpage, and a directed edge (i->j) means
that webpage i has a link to webpage j. Now given a webpage network graph G, you need to analyze the
importance of each webpage and give the top three webpage ids. G: The nodes are numbered from 1 to 20,
and the edges are: (1->5) (1->9) (2->6) (2->17) (3->8) (3->12) (4->2) (4->19) (5->14) (6->1) (7->10)
(7->3) (8->13) (9->4) (10->18) (11->7) (12->15) (13->20) (14->11) (15->2) (16->8) (16->5) (17->16)
(18->14) (19->6) (20->1) (5->7) (10->12) (15->19) (6->11) (8->2) (12->4) (14->9) (3->20) (7->15) (13-
>18) (9->17) (2->13) (11->16) (4->10).
------------------------------------------------------------------------------------------------
GraphWiz(LLaMA 2-7B): \n\n### The output is: ### The output is: The graph has 21 nodes and 21 edges, so
the number of edges is equal to the number of nodes. Therefore, the graph is connected and there is a path
between any two nodes. The graph is not bipartite, so there is no way to divide the nodes into two groups
such that no two nodes in the same group are directly connected. ✗

GraphWiz(LLaMA 2-7B-DPO): The graph has 21 nodes and 19 edges, and we can use a similar strategy as
the previous examples to find the top three nodes ... Finally, we can select node 6, which has the lowest
degree (one) among the remaining nodes. \nThis way, we obtain a valid topological sorting of the graph that
respects the direction of the edges. \n### The top three nodes are 1, 2, and 6. ✗

GraphAgent-Reasoner: To analyze the importance of each webpage in the given directed graph, we can use a
distributed algorithm based on the concept of PageRank. Below is the structured algorithm design:
### State:
1. `rank`: Represents the importance of the webpage, initialized to 1.0.
### Initialization
Step 1: Set `rank` to 1.0 for each node (webpage).
### Message:
1. `contribution`: Current rank of the node divided by its number of outgoing links.
Distributed_Algorithm_Designer: ### Send
Step 1: Calculate `contribution` as `rank / out_degree`, where `out_degree` is the total number of outgoing
links.
Step 2: Send `contribution` to each connected neighbor (Node A and Node B).
### Update
Step 1: Sum all received `contribution` values from messages.
Step 2: Update `rank` using the formula:
  - `new_rank = damping_factor * (sum_of_received_contributions) + (1 - damping_factor) * initial_rank`.
Step 3: Set the new `rank` as the current state.

<Distributed_Execution>

[RESULT]: Node: 1  State: 1. rank: 1.1194682167142727
Node: 2  State: 1. rank: 1.1624009025870747
...
Node: 19  State: 1. rank: 0.662517899831644        ✓
Node: 20  State: 1. rank: 0.5902983815488694

Master: {'thought': 'The PageRank values for each node have been calculated. The top three important
webpages are those with the highest ranks.', 'speak': 'The top three important webpages are 16, 14, and 5
based on their PageRank values.'}
```

Figure 5: The importance analysis in webpage network. While the GraphWiz fails due to incorrect graph assessments, GAR correctly uses the PageRank algorithm to identify nodes 16, 14, and 5 as the most important.

Although GraphWiz performed well on fine-tuned tasks, it exhibits severe overfitting when faced with real-world graph problems, failing to apply the graph reasoning knowledge learned during the fine-tuning phase. Since GraphWiz uses a consistent graph node description, the sentence "The nodes are numbered from 0 to ..." appears across all datasets during the mixed-task instruction tuning. When the actual problem has nodes numbered from 1 to 20, it still assumes the existence of node 0. As a result, both GraphWiz models first output that the graph has 21 nodes and an incorrect number of edges. Furthermore, neither of the two GraphWiz models recognizes that this is a problem associated with web page importance ranking. Instead, they approach it as the bipartite graph check or topological sort problems they had been fine-tuned on. Additionally, neither model generates an explicit and correct reasoning path. These observations indicate that there is still a significant gap between excelling in classic graph reasoning tasks and effectively solving real-world graph reasoning problems. In contrast, GAR correctly identifies that the problem should be solved using knowledge related to PageRank (Yang et al., 2024) and designs an algorithm that adhered to the distributed

paradigm (Note: the distributed algorithm library does not contain a PageRank algorithm template). GAR then assigns the algorithm to agent nodes for execution, ultimately obtaining the PageRank value for each node and arriving at the correct conclusion. Through the distributed paradigm, GAR effectively bridges the powerful knowledge learned by LLMs with the solving of real-world graph reasoning problems, which enables it to flexibly handle practical issues in a distributed manner. This case study demonstrates the feasibility of using GAR to solve real-world graph reasoning problems, indicating its substantial practical applicability and offering researchers and practitioners a powerful framework to address such tasks.

## 6 CONCLUSION

We first summarize three key issues faced by existing LLMs in graph reasoning tasks: limited graph scale, poor performance, and the lack of explicit reasoning paths. We then reflect on the limitations of a single LLM in addressing graph reasoning problems, such as the graph structures being too complex to memorize and understand and the overwhelming information in real-world graph reasoning scenarios. To address these challenges, we propose GraphAgent-Reasoner, a framework based on multi-agent collaboration to solve graph reasoning problems. This framework demonstrates superior accuracy and scalability, significantly surpassing existing closed-source and fine-tuned open-source models. Our experiments show its robust scalability, maintaining high accuracy on large graphs (up to 1,000 nodes). Our case study on webpage importance analysis further illustrates its capability to handle real-world graph reasoning problems. Future work will focus on designing more accurate and scalable LLM-based multi-agent graph reasoning frameworks, aiming to apply them to larger and more complex real-world reasoning scenarios.

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

## A  DISTRIBUTED ALGORITHMS UNDER THE DISTRIBUTED PARADIGM

### A.1  EXAMPLE OF DISTRIBUTED ALGORITHMS IN DISTRIBUTED ALGORITHM LIBRARY

**Shortest Path:** See Figure 6.

**Connectivity:** See Figure 7.

### A.2  EXAMPLE OF DISTRIBUTED ALGORITHMS DESIGNED BY THE MASTER LLM

**PageRank:** See Figure 8.

**Hamilton Path:** See Figure 9.

**Subgraph Matching:** See Figure 10.

## B  THE GRAPHINSTRUCT DATASET

The statistics and detailed information of GraphInstruct are shown in Table 3. Hamilton Path and Subgraph Matching are NP-complete problems.

**Hamilton Path Execution Example.**

Problem Description:

Determine whether or not there is a Hamiltonian path in an undirected graph. In an undirected graph, (i,j) means that node i and node j are connected with an undirected edge. Given a graph, you need to output Yes or No, indicating whether there is a Hamiltonian path in the graph. Q: The nodes are numbered from 0 to 5, and the edges are: (0, 3) (0, 1) (0, 2) (0, 4) (1, 5) (1, 4) (1, 2) (1, 3) (2, 4) (2, 5) (3, 5) (3, 4). Is there a Hamiltonian path in this graph?

Execution Process:

```
### State
1. `visited`: Boolean indicating if the node has been visited.
2. `path_length`: Integer representing the current length of the path.
3. `max_path_length`: Integer tracking the longest path found.
 ### Initialization
Step 1: Set `visited = False`, `path_length = 0`, and `max_path_length = 1` for all 
```

# Shortest Path

**State:**
1. `distance`: Known shortest distance from source node

**Initialization:**
For each node, the input is:
Node Id: <node_id>
---
### Initialization
Step 1: If Node Id is 14, set `distance` to 0; otherwise, set `distance` to infinity.

**Message:**
1. `new_distance`: Computed minimum distance from node 2 to the sending node.

**Send:**
For each node, the input is:
Node Id: <node_id>
State: 1. distance: <distance>
Neighbor Information:
Connected to:
Node A with weight <weight1>
Node B with weight <weight2>
---
### Send
Step 1: For each neighbor node, calculate `new_distance` = <distance> + <weight>
Step 2: Construct Message sent to Node xx: 1. new_distance: <new_distance>
### Example
## Input
Node Id: 5
State: 1. distance: 2
Neighbor Information:
Connected to:
Node A with weight 3
Node B with weight 5
## Process
For Node A with weight 3,
Step 1: calculate `new_distance` = 2 + 3 = 5
Step 2: construct Message sent to Node A: 1. new_distance: 5
For Node B with weight 5,
Step 1: calculate `new_distance` = 2 + 5 = 7
Step 2: construct Message sent to Node B: 1. new_distance: 7
## Output
Message sent to Node A: 1. new_distance: 5
Message sent to Node B: 1. new_distance: 7

**Update:**
For each node, the input is:
Node Id: <node_id>
State: 1. distance: <distance>
Received Messages:
Message: 1. new_distance: <new_distance>
Message: 1. new_distance: <new_distance>
---
### Update
Step 1: For each received message, check if `new_distance` is less than current `distance`.
Step 2: If it is, update `distance` to `new_distance`; otherwise, keep `distance` unchanged.
### Example
## Input
Node Id: 5
State: 1. distance: 15
Received Messages:
Message: 1. new_distance: 12
Message: 1. new_distance: 17
## Process
For Message: 1. new_distance: 12,
Step 1: `new_distance` = 12 is less than current `distance` = 15.
Step 2: update `distance` to 12. State: 1. distance: 12
For Message: 1. new_distance: 17
Step 1: `new_distance` = 17 is not less than current `distance` = 12.
Step 2: no update needed. State remains: 1. distance: 12
## Output
State: 1. distance: 12

Figure 6: Distributed algorithm for shortest path problem under the distributed paradigm.

```
                              Connectivity
State:
1. `Component_Id`: unique identifier for the connected component.
Message:
1. `Sender_Component_Id`: the current component identifier of the sender node.
Initialization:
For each node, the input is:
Node Id: <node_id>
---
### Initialization
Step 1: set Component_Id = <node_id>
Send:
For each node, the input is:
Node Id: <node_id>
State: 1. Component_Id: <component_id>
Neighbor Information:
Connected to:
Node A
Node B
---
### Send
Step 1: Construct Message sent to Node xx: 1. Component_Id: <component_id>
### Example
## Input
For each node, the input is:
Node Id: 5
State: 1. Component_Id: 5
Neighbor Information:
Connected to:
Node A
Node B
## Process
For Node A,
Step 1: Construct Message sent to Node A: 1. Component_Id: 5
For Node B,
Step 1: Construct Message sent to Node B: 1. Component_Id: 5
## Output
Message sent to Node A: 1. Component_Id: 5
Message sent to Node B: 1. Component_Id: 5
Update:
For each node, the input is:
Node Id: <node_id>
State: 1. Component_Id: <component_id>
Receive Messages:
Message: 1. Sender_Component_Id: <sender_component_id>
Message: 1. Sender_Component_Id: <sender_component_id>
---
### Update
Step 1: For each received message, check if <sender_component_id> is less than current
<component_id>
Step 2: If it is, update <component_id> to <sender_component_id>; otherwise, keep
<component_id> unchanged.
### Example
## Input
For each node, the input is:
Node Id: 5
State: 1. Component_Id: 5
Receive Messages:
Message: 1. Sender_Component_Id: 2
Message: 1. Sender_Component_Id: 7
## Process
For Message: 1. Sender_Component_Id: 2,
Step 1: `Sender_Component_Id` = 2 is less than current `Component_Id` = 5.
Step 2: update `Component_Id` to 2. State: 1. Component_Id: 2
For Message: 1. Sender_Component_Id: 7,
Step 1: `Sender_Component_Id` = 7 is not less than current `Component_Id` = 2.
Step 2: no update needed. State remains: 1. Component_Id: 2
## Output
State: 1. Component_Id: 2'''
```

Figure 7: Distributed algorithm for connectivity problem under the distributed paradigm.

```
                              PageRank
State:
1. `rank`: Represents the importance of the webpage, initialized to 1.0.
Initialization:
Step 1: Set `rank` to 1.0 for each node (webpage).
Step 2: (No additional steps required).
Message:
1. `contribution`: Current rank of the node divided by its number of outgoing links.
Send:
Step 1: Calculate `contribution` as `rank / out_degree`, where `out_degree` is the total
number of outgoing links.
Step 2: Send `contribution` to each connected neighbor (Node A and Node B).
### Example
## Input
Node Id: 1
State: 1. rank: 2.0
Neighbor Information:
Connected to:
Node A
Node B
## Process
1. Calculate `out_degree` for Node 1:
   - Outgoing links = 2 (to Node A and Node B).
2. Calculate `contribution`:
   - `contribution = rank / out_degree = 2.0 / 2 = 1.0`.
3. Send `contribution` to neighbors:
   - Message to Node A: `contribution = 1.0`.
   - Message to Node B: `contribution = 1.0`.
## Output
Message sent to Node A: 1. contribution: 1.0
Message sent to Node B: 1. contribution: 1.0
Update:
Step 1: Sum all received `contribution` values from messages.
Step 2: Update `rank` using the formula:
   - `new_rank = damping_factor * (sum_of_received_contributions) + (1 - damping_factor) *
initial_rank`.
Step 3: Set the new `rank` as the current state.
### Example
## Input
Node Id: 2
State: 1. rank: 1.5
Received Messages:
Message: 1. contribution: 0.8
Message: 2. contribution: 1.2
## Process
1. Sum all received `contribution` values:
   - `sum_of_received_contributions = 0.8 + 1.2 = 2.0`.
2. Update `rank` using the formula:
   - `new_rank = damping_factor * (sum_of_received_contributions) + (1 - damping_factor) *
initial_rank`.
   - Assuming `damping_factor = 0.85`:
   - `new_rank = 0.85 * 2.0 + 0.15 * 1.5 = 1.7 + 0.225 = 1.925`.
3. Set the new `rank` as the current state:
   - `rank = 1.925`.
## Output
State: 1. rank: 1.925
```

Figure 8: Distributed algorithm for pagerank calculation under the distributed paradigm.

864
865
866
867
868
869
870
871
872
873
874
875
876
877
878
879
880
881
882
883
884
885
886

## Hamilton Path

**State:**
1. 'isInPath': boolean (True if the node is part of the Hamiltonian path, False otherwise);
2. 'pathLength': integer (length of the path ending at this node).

**Initialization:**
Step 1: Each node initializes `isInPath` as `False` and `pathLength` as `1`.
Step 2: Node 0 initializes `isInPath` as `True` and `pathLength` as `1`.

**Message:**
1. 'pathLength': integer (current length of the path from the sender node); 2. 'isInPath': boolean (True if the sender node is part of the Hamiltonian path).

**Send:**
Step 1: For each neighbor (Node A, Node B), if `isInPath` is `True`, send messages containing:
- `pathLength`: current `pathLength` + 1
- `isInPath`: current `isInPath`.
Step 2: If `isInPath` is `False`, do not send messages.

**Update:**
Step 1: For each received message, check if `isInPath` from the message is `True`.
Step 2: If `isInPath` is `True` and the `pathLength` from the message is greater than the current `pathLength`, update the state:
- Set `pathLength` to the maximum of the current `pathLength` and the received `pathLength` + 1.
- Set `isInPath` to `True` (if not already).
Step 3: If no messages resulted in an update, retain the current state.

Figure 9: Distributed algorithm for hamilton path problem under the distributed paradigm.

887
888
889
890
891
892
893
894
895
896
897
898
899
900
901
902
903
904
905
906
907
908
909
910
911
912
913
914
915
916
917

## Subgraph Matching

**State:**
1. **NodeMatch**: A list of potential matches for nodes in the subgraph G'.
2. **MatchedFlag**: A boolean indicating if the node is currently matched to a node in the subgraph.

**Initialization:**
Step 1: Each node in graph G initializes its **NodeMatch** as an empty list and **MatchedFlag** as false.
Step 2: Each node in subgraph G' initializes its unique identifier and sets **MatchedFlag** as false.

**Message:**
1. **MatchInfo**: Contains the identifier of the sending node and its list of current potential matches.
2. **EdgeConnection**: Indicates the directed neighbors that the sending node is connected to.

**Send:**
Step 1: Construct **MatchInfo** message containing the node identifier and its list of potential matches.
Step 2: Include **EdgeConnection** message detailing connections to both Node A and Node B.
Step 3: Send the constructed messages to both connected neighbors.

**Update:**
Step 1: For each received **MatchInfo** message, check if the identifier matches any node in the state of the subgraph G'. If a match is found, update **NodeMatch** to include the new match and set **MatchedFlag** to true.
Step 2: For each received **EdgeConnection** message, update the list of potential neighbors or confirm connectivity with received nodes, ensuring to track relationships needed for further matching.
Step 3: If no new matches are found and the **MatchedFlag** remains false, retain the current state while preparing for potential future updates.

Figure 10: Distributed algorithm for subgraph matching problem under the distributed paradigm.

Table 3: The detailed information of GraphInstruct dataset.

| Problem | Definition | Node Range | Test Size |
|---------|-----------|------------|-----------|
| Cycle Detection | Detect if a given graph $\mathcal{G}$ contains any cycles. | [2, 100] | 400 |
| Connectivity | Assess if two nodes $u$ and $v$ in a given graph $\mathcal{G}$ are connected via a path. | [2, 100] | 400 |
| Bipartite Graph Check | Judge if a given graph $\mathcal{G}$ is bipartite. | [2, 100] | 400 |
| Topological Sort | Find a topological ordering of vertices in a directed acyclic graph $\mathcal{G}$. | [2, 50] | 400 |
| Shortest Path | Compute the shortest path between two specific nodes $u$ and $v$ in a given graph $\mathcal{G}$. | [2, 100] | 400 |
| Maximum Triangle Sum | Find the maximum sum of weights for any connected triplet of vertices in a given graph $\mathcal{G}$. | [2, 25] | 400 |
| Maximum Flow | Calculate the maximum flow from a source node $s$ to a sink node $t$ in a directed graph $\mathcal{G}$. | [2, 50] | 400 |
| Hamilton Path | Determine if a given graph $\mathcal{G}$ has a Hamiltonian path that visits each vertex exactly once. | [2, 50] | 400 |
| Subgraph Matching | Verify if there exists a subgraph in $\mathcal{G}$ that is isomorphic to a given graph $\mathcal{G}'$. | [2, 30] | 400 |

```
Step 2: For the initial node (e.g., Node Id: 0), set `visited = True` and
`path_length = 1`.
Distributed_Algorithm_Designer: ### Message
1. `path_length`: The current path length from the sender.
2. `max_path_length`: The maximum path length known to the sender.
3. `visited_flag`: Boolean indicating whether the sender has visited the node.
### Send
Step 1: For each neighbor (Node A, Node B), if `visited` is `False`, construct the me
- Message to Node A: `(Node Id: <node_id>, path_length + 1, max(max_path_length, path
- Message to Node B: `(Node Id: <node_id>, path_length + 1, max(max_path_length, path
Step 2: Send the constructed messages to both neighbors.
### Example
## Input
Node Id: 0
State:
1. visited: False
2. path_length: 2
3. max_path_length: 3
Neighbor Information:
Connected to:
Node A
Node B
## Process
Step 1: Check neighbors Node A and Node B. Since `visited` is
`False`, proceed to construct messages.
- For Node A:
  - Calculate new `path_length`: `2 + 1 = 3`
  - Calculate `max_path_length`: `max(3, 2 + 1) = 3`
- For Node B:
  - Same calculations apply as for Node A.
Step 2: Construct messages:
- Message to Node A: `(Node Id: 0, 3, 3)`
- Message to Node B: `(Node Id: 0, 3, 3)`
## Output
Message sent to Node A:
1. path_length: 3
2. max_path_length: 3
3. visited_flag: False
```

```
Message sent to Node B:
1. path_length: 3
2. max_path_length: 3
3. visited_flag: False
### Update
Step 1: For each received message, check the `visited_flag`. If it is
`False`, proceed to the next steps.
Step 2: Update `visited` to `True` for the current node.
Step 3: Update `path_length` to the maximum of its current value and
the received `path_length` from the message, incrementing by 1 if the
message indicates a potential continuation of the path.
Step 4: Update `max_path_length` to the maximum of its current value
and the received `max_path_length` from the message.
Step 5: If any messages were processed, trigger a re-evaluation of
outgoing messages to neighbors, if applicable.
### Example
## Input
Node Id: 1
State:
1. visited: False
2. path_length: 1
3. max_path_length: 2
Received Messages:
Message 1:
1. path_length: 2
2. max_path_length: 3
3. visited_flag: False
Message 2:
1. path_length: 3
2. max_path_length: 4
3. visited_flag: True
## Process
Step 1: Process Message 1. Since `visited_flag` is `False`, proceed:
- Update `visited` to `True`.
- Update `path_length`: `max(1, 2) + 1 = 3`.
- Update `max_path_length`: `max(2, 3) = 3`.
Step 2: Process Message 2. Since `visited_flag` is `True`, ignore this
message and do not make any changes.
## Output
State:
1. visited: True
2. path_length: 3
3. max_path_length: 3
### Termination
The algorithm continues until:
- A node reaches a `path_length` equal to the total number of nodes
(indicating a Hamiltonian path exists).
- Or until a predefined number of iterations equal to the total number
- of nodes (`N`) is reached without discovering a Hamiltonian path.
Master: #################### ITER 3, STEP 1: REASONING ####################
Master: {'thought': "The algorithm is designed to check for a
Hamiltonian path. Now let's execute it to see if such a path exists.",
'speak': "The algorithm is ready. Let's execute it to determine if there is
a Hamiltonian path.", 'function': [{'name': 'Distributed_Execution',
'arguments': {}}]}
Master: #################### ITER 3, STEP 2: ACTING ####################
0: State: 1. visited: True 2. path_length: 1 3. max_path_length: 1
2: State: 1. visited: False 2. path_length: 0 3. max_path_length: 1
5: State: 1. visited: False 2. path_length: 0 3. max_path_length: 1
```

```
3: State: 1. visited: False 2. path_length: 0 3. max_path_length: 1
4: State: 1. visited: False 2. path_length: 0 3. max_path_length: 1
1: State: 1. visited: False 2. path_length: 0 3. max_path_length: 1
Node 5 Send Message to Node 1: 1. path_length: 1 2. max_path_length:
1 3. visited_flag: False
Node 5 Send Message to Node 2: 1. path_length: 1 2. max_path_length:
1 3. visited_flag: False
Node 5 Send Message to Node 3: 1. path_length: 1 2. max_path_length:
1 3. visited_flag: False
...
All agents' state unchanged, terminating early...
>>> END
system: 1. Execute function Distributed_Execution
    [ARGUMENTS]:

    [STATUS]: SUCCESS
    [RESULT]: Node: 0  State: 1. visited: True 2. path_length: 5 3. max_path_length: 1
Node: 1  State: 1. visited: True 2. path_length: 2 3. max_path_length: 1
Node: 2  State: 1. visited: True 2. path_length: 4 3. max_path_length: 3
Node: 3  State: 1. visited: True 2. path_length: 5 3. max_path_length: 3
Node: 4  State: 1. visited: True 2. path_length: 4 3. max_path_length: 3
Node: 5  State: 1. visited: True 2. path_length: 5 3. max_path_length: 3

Master: #################### ITER 4, STEP 1: REASONING ####################
Master: {'thought': 'Since the maximum path length found is 5, which is
less than the total number of nodes (6), there is no Hamiltonian path in
the graph.', 'speak': 'No, there is no Hamiltonian path in this graph.',
'function': []}
```

## C   EXECUTION EXAMPLES OF GRAPHAGENT-REASONER

**Shortest Path Execution Example.**

Problem Description:

Find the shortest distance from a source node to other nodes in an undirected graph. In an undirected graph, (i,j,k) means that node i and node j are connected with an undirected edge with weight k. The graph has 8 nodes, and the edges are: (0,7,9) (0,1,7) (0,4,9) (1,7,1) (2,7,7) (2,6,5) (2,5,8) (3,5,9) (3,4,8) (3,6,1) (4,7,7) (4,5,6) (5,7,6). Give the weight of the shortest distance from node 1 to other node.

Execution Process:

```
Master: #################### ITER 1, STEP 1: REASONING ####################
Master: {'thought': 'I need to find the shortest distances from node 1 to all
other nodes in the given undirected graph. First, I will build the graph using
the provided edges and then design a distributed algorithm to find the
shortest paths.', 'speak': 'Building the graph and designing the distributed
algorithm to find the shortest distances from node 1.', 'function': '[{"name":
"Build_Graph", "arguments": {"Edges": [[0, 7, 9], [0, 1, 7], [0, 4, 9], [1, 7, 1], [2,
7, 7], [2, 6, 5], [2, 5, 8], [3, 5, 9], [3, 4, 8], [3, 6, 1], [4, 7, 7], [4, 5, 6],
"Nodes_Num": 8, "Directed": false}}]'}
Master: #################### ITER 1, STEP 2: ACTING ####################
>>>     Edges: [[0, 7, 9], [0, 1, 7], [0, 4, 9], [1, 7, 1], [2, 7...
>>>     Nodes_Num: 8
>>>     Directed: False
>>> END
system: 1. Execute function Build_Graph
    [ARGUMENTS]:
        Edges: [[0, 7, 9], [0, 1, 7], [0, 4, 9], [1, 7, 1], [2, 7, 7], [2, 6, 5], [2,
```

```
1080            5, 9], [3, 4, 8], [3, 6, 1], [4, 7, 7], [4, 5, 6], [5, 7, 6]]
1081                    Nodes_Num: 8
1082                    Directed: False
1083      [STATUS]: SUCCESS
1084  Master: #################### ITER 2, STEP 1: REASONING ####################
1085  Master: {'thought': 'The graph has been successfully built. Now, I will design
1086  a distributed algorithm to find the shortest distances from node 1 to all
1087  other nodes.', 'speak': 'Designing a distributed algorithm to find the
1088  shortest distances from node 1.', 'function': '[{"name":
1089  "Design_Distributed_Algorithm", "arguments": {"Problem_Str": "Find the
1090  shortest distance from node 1 to all other nodes in an undirected graph
1091  with 8 nodes and given edges."}}]'}
1092  Master: #################### ITER 2, STEP 2: ACTING ####################
1093  >>> END
1094  Master: #################### ITER 3, STEP 1: REASONING ####################
1095  Master: {'thought': 'The distributed algorithm has been successfully
1096  designed. Now, I will execute the algorithm to find the shortest distances
1097  from node 1 to all other nodes.', 'speak': 'Executing the distributed
1098  algorithm to calculate the shortest distances from node 1.', 'function':
1099  '[{"name": "Distributed_Execution", "arguments": {}}]'}
1100  Master: #################### ITER 3, STEP 2: ACTING ####################
1101  Initialization:
1102  1: State: 1. distance: 0
1103  0: State: 1. distance: \infinity
1104  2: State: 1. distance: \infinity
1105  4: State: 1. distance: \infinity
1106  5: State: 1. distance: \infinity
1107  7: State: 1. distance: \infinity
1108  3: State: 1. distance: \infinity
1109  6: State: 1. distance: \infinity
1110  Node 6 Send Message to Node 2: 1. new_distance: \infinity
1111  Node 6 Send Message to Node 3: 1. new_distance: \infinity
1112  Node 1 Send Message to Node 0: 1. new_distance: 7
1113  Node 1 Send Message to Node 7: 1. new_distance: 1
1114  Node 3 Send Message to Node 5: 1. new_distance: \infinity
1115  ....
1116  All agents' state unchanged, terminating early...
1117  >>> END
1118  system: 1. Execute function Distributed_Execution
1119      [ARGUMENTS]:
1120
1121      [STATUS]: SUCCESS
1122      [RESULT]: Node: 0   State: 1. distance: 7
1123  Node: 1   State: 1. distance: 0
1124  Node: 2   State: 1. distance: 8
1125  Node: 3   State: 1. distance: 14
1126  Node: 4   State: 1. distance: 8
1127  Node: 5   State: 1. distance: 7
1128  Node: 6   State: 1. distance: 13
1129  Node: 7   State: 1. distance: 1
1130
1131  Master: #################### ITER 4, STEP 1: REASONING ####################
1132  The shortest distances from node 1 are as follows: Node 0: 7, Node 1: 0,
1133  Node 2: 8, Node 3: 14, Node 4: 8, Node 5: 7, Node 6: 13, Node 7: 1.Master:
       {'thought': 'The distributed algorithm has been successfully executed, and I
       have obtained the shortest distances from node 1 to all other nodes.',
       'speak': 'The shortest distances from node 1 are as follows: Node 0: 7, Node
       1: 0, Node 2: 8, Node 3: 14, Node 4: 8, Node 5: 7, Node 6: 13, Node 7: 1.',
       'function': []}
```

