# OpenReview forum: "Scalable and Accurate Graph Reasoning with LLM-based Multi-Agents"
_ICLR.cc/2025/Conference — ICLR 2025 Conference Withdrawn Submission_

### Official Review · Reviewer_Me6P · 2024-10-27

**Soundness:** 1
**Presentation:** 2
**Contribution:** 2
**Rating:** 3
**Confidence:** 5

**Summary:**

This paper presents a multi-agent framework named GraphAgent-Reasoner for solving graph reasoning problems. Inspired by distributed algorithms, the framework provides detailed algorithmic processing steps for several fundamental graph theory tasks that can be processed in a distributed manner, and presents a general multi-agent distributed strategy for handling graph reasoning problems. The paper uses part of the GraphWiz dataset and compares against GraphWiz-DPO, demonstrating that proposed framework and algorithms can achieve good performance on certain fundamental graph theory problems.

**Strengths:**

1. This paper analyzes why a single LLM cannot effectively handle large-scale graph analysis problems from a memory perspective, offering a novel insight.
2. The paper constructs a multi-agent framework for distributed graph reasoning, which can solve several fundamental graph theory problems mentioned in the paper. The results demonstrate that GraphAgent-Reasoner can achieve good performance on certain problems.

**Weaknesses:**

1. The paper claims to solve complex graph reasoning problems, but only 6 out of 9 problem types in GraphWiz dataset were evaluated. Of the remaining three categories, one (maximum flow) was marked as Medium by GraphWiz authors, and two (hamilton path and subgraph matching) were marked as Hard. No corresponding results were provided. If distributed algorithms are not a universal graph reasoning method, this paper appears to be over-claiming.
2. According to the algorithms in appendix, I think the method may suffer from high computational cost. Comparisons about reasoning time, token consumption, and corresponding expenses are needed.
3. There are many issues about experiment settings and presentation, making me feel the paper was not well-prepared. See the questions below.

**Questions:**

1. In the appendix, Table 3 mentions that GraphInstruct dataset has 9 problem types and has maximum flow, hamilton path and subgraph matching problems. The authors only provide experimental results for 6 types of problems. What about the results for the other problem types? Without investigating these complex problems, the claimed contributions about their framework in solving complex graph reasoning problems appear to be problematic.

2. It appears the distributed algorithms suffers from high computational complexity. Can you provide a table showing how the number of agents needed, reasoning time, token consumption, and corresponding expenses with increasing graph size? Comparisons with baselines are needed.

3. Did the authors conduct repeated experiments in the reported results? No descriptions about repeated experiments were given.

4. In the appendix (P13, P18, P19 and P20), a Hamilton Path Execution Example is provided.  However, I found the execution result by GraphAgent-Reasoner seems to be wrong. I also test this problem with GPT4-turbo and obtain a correct path, making me doubt about the soundness of this work.

5. There is a distributed algorithm for PageRank in the appendix, but no relevant quantity experiments are given.

6. In Figure 4(a), why GraphAgent and GraphWiz-DPO’s performance is better with a size of 80 than with 60?

---

### Official Review · Reviewer_HrHA · 2024-11-02

**Soundness:** 3
**Presentation:** 4
**Contribution:** 4
**Rating:** 5
**Confidence:** 5

**Summary:**

This paper proposes GraphAgent-Reasoner (GAR), a framework that leverages a multi-agent collaboration strategy for precise and scalable graph reasoning. It consists of four key steps: Graph Construction, Algorithm Establishing, Distributed Execution, and Master Summarization. Extensive experiments on graph reasoning tasks from GraphInstruct indicate that GAR significantly improves accuracy while preserving robust scalability, surpassing both closed-source and fine-tuned open-source models.

**Strengths:**

1. This paper introduces a fine-tuning-free approach based on multi-agent collaboration to decompose complex graph reasoning problems into smaller, node-centric tasks, effectively improving accuracy while ensuring scalability.
2. This paper illustrates the inherent limitations of single LLMs in graph reasoning tasks in detail. Moreover, extensive experimental results show that the proposed GAR with multi-agent collaboration consistently outperforms all baselines, achieving near-perfect accuracy on polynomial-time graph reasoning tasks.
3. The paper is well-organized and written, with a clear problem setup, related work, methodology, and experimental evaluation.

**Weaknesses:**

1. The paper does not provide comprehensive ablation and hyperparameter studies, such as evaluating the impact of using different LLMs, varying the maximum number of iterations, and the effect of incorporating a distributed algorithm library.
2. The paper lacks an analysis of the computational complexity, efficiency, and resource requirements of the proposed GAR, including the training/inference time, memory usage, and API costs, which would help assess its scalability and practical applicability.
3. The paper misses some important baselines, particularly the single LLM utilizing various advanced prompting techniques such as Chain-of-Thought (CoT) [1], Tree of Thought (ToT) [2], and Iteration of Thought (IoT) [3]. Comparing the proposed GAR with these baselines would further demonstrate its effectiveness and scalability.
4. Several typos are present in the paper. For example, In line 80, "Reasoner(GAR) framework, ..." lacks a space after "Reasoner". In line 173, it would be beneficial to spell out "TAG" as "Text-Attributed Graph" for clarity. In the caption of Figure 4 (line 374), there are missing spaces after "GPT4" and "GraphWiz".

**Reference**
[1] Chain-of-thought prompting elicits reasoning in large language models, NeurIPS, 2022.
[2] Tree of Thoughts: Deliberate Problem Solving with Large Language Models, NeurIPS, 2023.
[3] Iteration of Thought: Leveraging Inner Dialogue for Autonomous Large Language Model Reasoning, 2024.

**Questions:**

1. In lines 301-302, it states, "Communication will continue until the maximum number of iterations is reached or the termination condition is met." Could the authors clarify the specific criteria for the termination condition?
2. Could the authors verify the performance of GAR in more complex graph reasoning tasks from GraphInstruct, such as Hamilton Path and Subgraph Matching, to provide a more comprehensive evaluation of its accuracy and scalability?
3. The paper seems to underutilize the reasoning and generative capabilities of LLMs, which could potentially improve GAR’s interpretability and effectiveness in graph reasoning tasks. Could the authors clarify this?

---

### Official Review · Reviewer_EGws · 2024-11-03

**Soundness:** 3
**Presentation:** 3
**Contribution:** 2
**Rating:** 3
**Confidence:** 5

**Summary:**

The paper presents GraphAgent-Reasoner (GAR), a scalable and accurate framework for graph reasoning that leverages Large Language Models (LLMs) through a multi-agent collaboration strategy. By decomposing complex graph problems into node-centric tasks assigned to individual agents, GAR reduces the processing load on a single LLM, enabling the system to scale efficiently to graphs with over 1,000 nodes without a loss in accuracy. Unlike previous methods limited by LLMs' handling of long inputs, GAR achieves near-perfect accuracy on polynomial-time tasks, significantly outperforming existing models

**Strengths:**

1. The paper is well-written and easy to follow.

2. This paper identifies a critical issue that larger graph size leads to performance drop in graph reasoning task.

3. The proposed framework achieves promising results compared to baselines, especially in larger graphs.

**Weaknesses:**

1. Though the performance in large-scale graph reasoning tasks is promising, the overall novelty of this framework is limited. Multi-agent systems have been proposed in many other domains, and this work seems to simply adopt the idea for the graph reasoning task without a specific design tailored to it.

2. The implementation details for Experiment 2 are unclear. How is the dataset constructed, and what do the 20 samples look like? I would suggest that the authors add more details to improve the soundness of the experiments.

3. The experiments are not comprehensive. Although all experiment results look favorable according to the paper, I am wondering about the performance limits of this framework. At what graph size does the performance begin to drop?

4. Though the authors claim that they aim to address the challenges of large graphs, I still believe this issue is not well-addressed. The largest graph size in this paper is 1,000, yet many real-world applications involve graphs much larger than 1,000 nodes. How can this framework be adapted to such scenarios? Especially, how could we optimize efficiency to handle these larger graphs? It seems that the computational cost is very large by modeling each node as an agent, which makes it not scalable to really large graphs. What is the computational cost?

5. Many important details are missing, such as how the messages are prepared and sent and how each agent updates their state. Though the authors provide some examples in the appendix, the paper lacks a description of the principled design of these important components in the main content.

6.	The proposed GAR is given some distributional algorithms for solving graph problems. For a fair comparison, the baselines should also be given a list of graph algorithms, such as algorithms for finding the shortest path, to solve the given tasks. Could you please clarify if these baselines are given such algorithms?

**Questions:**

Please see the above weaknesses

---

### Note · Authors · 2024-11-18

**Comment:**

We sincerely thank the reviewers for their invaluable comments. We will work to improve the paper further and submit it elsewhere.

**Withdrawal Confirmation:**

I have read and agree with the venue's withdrawal policy on behalf of myself and my co-authors.